

# Benzene and Toluene in the surface air of North Eurasia from TROICA-12 campaign along the Trans-Siberian railway

Andrey I. Skorokhod[1], Elena V. Berezina[1], Konstantin B. Moiseenko[1], Nikolai F. Elansky[1], Igor B. Belikov[1]

[1]A.M. Obukhov Institute of Atmospheric Physics, Russian Academy of Sciences, Moscow, 119017, Russia

*Correspondence to*: Andrey I. Skorokhod (askorokhod@mail.ru) and Elena V. Berezina (e_berezina_83@mail.ru)

**Abstract.** Volatile organic compounds (VOCs) were measured by proton transfer reaction – mass spectrometry (PTR-MS) on a mobile laboratory in a transcontinental TROICA-12 (21.07.2008 – 04.08.2008) campaign along the Trans-Siberian railway from Moscow to Vladivostok. Surface concentrations of benzene ($C_6H_6$) and toluene ($C_7H_8$) along with non-methane hydrocarbons (NMHCs), CO, $O_3$, $SO_2$, NO, $NO_2$ and meteorology are analyzed in this study to identify the main sources of benzene and toluene along the Trans-Siberian railway. The most measurements in the TROICA-12 campaign were conducted under low-wind/stagnant conditions in moderately (~78% of measurements) to weakly polluted (~20% of measurements) air directly affected by regional anthropogenic sources adjacent to the railroad. Only 2% of measurements were identified as characteristic of highly polluted urban atmosphere. Maximum values of benzene and toluene during the campaign reached 36.5 ppb and 45.6 ppb, correspondingly, which is significantly less than their one-time maximum permissible concentrations (94 and 159 ppb for benzene and toluene, correspondingly). About 90% of benzene and 65% of toluene content is attributed to motor vehicle transport and 10% and 20%, correspondingly, provided by the other local and regional-scale sources. The highest average concentrations of benzene and toluene are measured in the industrial regions of the European Russia (up to 0.3 and 0.4 ppb for benzene and toluene, correspondingly) and south Siberia (up to 0.2 and 0.4 ppb for benzene and toluene, correspondingly). Total contribution of benzene and toluene to photochemical ozone production along the Trans-Siberian railway is about 16% compared to the most abundant biogenic VOC – isoprene. This contribution, however, is found to be substantially higher (up to 60–70%) in urbanized areas along the railroad suggesting important role of anthropogenic pollutant sources in regional ozone photochemistry and air quality.



# 1 Introduction

Aromatic compounds is an important class of hydrocarbons taking a significant part in the formation of tropospheric ozone and the secondary pollutants, including organic aerosols, that can lead to photochemical smog (Wayne, 2000; Baltaretu et al, 2009). Being toxic and carcinogenic, these gases, even without chemical transformation, have multiple impacts on the environment and human health (Derwent et al., 2000; Calvert et al., 2002; Lippmann, 2009). Thus, it is very important to study the role of aromatic compounds in air pollution for improving air quality.

A dominant source of aromatic compounds, particularly in urban environments, is automobile emissions with significant emissions being also from their use as solvents in various manufacturing processes (Wayne, 2000). Aromatic compounds can make up to 30-40% of the total nonmethane hydrocarbon (NMHC) content in the atmosphere, and be responsible for about 40% of the ozone producing potential of all NMHCs (Derwent et al., 2000; Calvert et al., 2002; Mugica et al., 2003; North et al., 2010).

In anthropogenically loaded regions, the most abundant aromatic compounds usually include benzene and toluene, which we focused our effort on in this study. They are often associated with air emissions from petroleum production industries (motor vehicle exhaust, incomplete combustion of fossil fuels, oil and gas service stations and other industrial and human activities). Although, Filella et al., 2006, White et al., 2009 and Parra et al., 2006 point out on the biogenic sources of benzene and toluene in summer rural conditions as well.

Many studies on benzene and toluene in urban and rural atmosphere have been carried out around the world and presented, for example, in (Keymeulen et al., 2001; Barletta et al., 2005; Parra et al., 2006; Velasko et al., 2007; Tivary et al., 2010; Miller et al., 2011; Civan et al., 2012; Chaudhary et al., 2012; Shaw et al., 2015). However, there is still a substantial lack of information on benzene and toluene abundance and spatial localization in remote areas of the North Eurasia. This gap is partially filled by the TROICA-12 campaign on a mobile carriage-laboratory in summer 2008 during which a complex measurements of surface air chemical composition were conducted along the Trans-Siberian railway from Moscow to Vladivostok (~ 9300 km long).

This study aims to investigate variation of benzene and toluene in the surface air over Russia from PTR-MS measurements to determine their characteristic levels in urban, rural, and remote atmosphere in different geographic regions with respect to their major regional sources including large towns and industrial areas along the railroad. The relative importance of benzene and toluene emissions in photochemical near-surface ozone production is then assessed with respect to a major biogenic ozone precursor, isoprene. (Chameides et al., 1992; Geng et al., 2011).

# 2 Methods

## 2.1 TROICA experiments



TROICA experiments over Russia on a mobile laboratory have been carried out regularly since 1995 (Elansky et al., 2009). About 10 inorganic compounds (ozone, nitric oxides, carbon oxides, methane, sulphur dioxide etc.) as well as aerosols and meteorology (air temperature, pressure and humidity, solar radiation, vertical temperature profiles, wind speed and direction (at stops)) were measured continuously and simultaneously by a specially constructed automated system. The system was

built on a railway carriage with air inlets at the height of about 4 m above the ground. VOC concentrations have been measured routinely since 2008 (TROICA-12, -13 and -14 campaigns). The TROICA carriage–laboratory is equipped in accordance with the measurement requirements of the Global Atmospheric Watch (WMO), and is located just after the electric locomotive to minimize various effects of near-surface air perturbations due to moving train. The possible impact of oncoming trains, e.g. those transporting oil products, on the measurements has been removed through the respective time

filtering of the original data. We also note that this impact is expected to be generally non-significant as demonstrated previously in (Krutzen et al., 1996; Elansky et al., 2000).

In present study, the data from the summer experiment TROICA-12 (21.07.2008 – 04.08.2008) along the Trans-Siberian railway (Fig.1) are analyzed. The train covers the total length of the route from Moscow to Vladivostok (~ 9288 km) for approximately 6 days, so the total duration of a single campaign is about two weeks. (Henceforth, we denote forward path

from Moscow to Vladivostok, and return path from Vladivostok to Moscow, as east and west segments of the whole route, correspondingly).

Undoubtedly, the results of the observations at each particular location performed from the moving carriage are strongly influenced by specific weather conditions (synoptic patterns), as well as complex interplay of local pollution sources and atmospheric transport by turbulent eddies on a variety of scales, the latter being considered as a source of random noise in

the measurement data. Generally, straightforward quantification of the effects of atmospheric dilution and absolute strength of the associated nearby emission sources is inhibited in data analyses. Yet, the passage of each location twice (in the forward and return paths of the TROICA campaign) allows for some qualitative assessment of the impact of weather conditions, since the time between the two subsequent measurements is of the order of week which is comparable to the characteristic time of the boundary layer ventilation in mid-latitudes.

Various types of meteorological conditions along the railroad can be generalized into three distinctive weather patterns changing each other when traversing mountain area adjacent to Baikal Lake (~110° E) in east and west segments. There was clear and warm (>20° C at noon) weather observed on the route from Moscow to Baikal, hot weather (>24° C at noon) observed between Baikal Lake area and Vladivostok (east and west segments), and cool (daytime temperatures of 15 – 20° C) and rainy weather observed between Baikal Lake and Moscow on the return west segment of the route. Nighttime surface

temperature inversions and stagnant air conditions were common for the east segment, although light winds were typical for the both east and west segments of TROICA. This feature is clearly seen on the Fig.1 on which 2-day back trajectories along the TROICA route calculated with the use of NOAA HIGHSPLIT model [Stein et al., 2015; Rolph, 2016] based on 3D wind fields are shown as averages of corresponding ensembles of trajectories originated within a height range from 0 – 400 m a.g.l. According to the figure, the characteristic distances of transport within a planetary boundary layer does not exceed 500



– 600 km in the two days preceding measurement time, so the measured chemical composition of the respective air masses can be considered as representative of the cumulative impact of pollutant sources at local to regional scale. Relatively low wind speeds and short chemical lifetimes of the analyzed species (~12 days for benzene and ~2-4 days for toluene) support a regional approach to data analyzes implemented in present study (see Section 3) to quantify impact of various anthropogenic

sources (large towns and industrial facilities) localized primarily along the railroad.  The exception is the Far East Region where regional advection by southerly winds may contribute to measured pollutant concentrations from highly urbanized areas of north-east China.

## 2.2 VOC measurements

VOCs concentration was measured by a Compact Proton Transfer Reaction Mass Spectrometer (Compact PTR-MS) from

Ionicon Corp., Austria. The main advantage of the Compact PTR-MS is its portability, which allows its using for continuous measurements on a mobile moving platform.

In the PTR-MS instrument, ambient air is continuously pumped through a drift tube reactor and the VOCs in the sample are ionized in proton transfer reactions with hydronium ions (H3O+). The reagent and product ions are selected according to their masses by a quadrupole mass spectrometer and detected by a secondary electron multiplier (Lindinger et al., 1998a).

The Compact PTR-MS measurement range depends on the substances measured, integration time and system set-up. Its detection limit for the investigated VOCs is in order of pptv. The PTR-MS drift tube was operated at 2 mbar and 40 °C with a drift voltage of 600 V. The detailed information about the instrument and its application in atmospheric gases measurements is presented in (Fortner, 2006).

The ions associated with benzene (m79) and toluene (m93) are free from significant interferences of other VOC species

(Warneke et al., 2003).  In the case of isoprene (m69), other aldehydes and ketones, which are known to be detectable at this mass (Warneke et al., 2003; Fortner et al., 2009), however, isoprene was often found as the dominant species at mass 69 within various kinds of air masses (de Gouw and Warneke, 2007). Nevertheless, the isoprene signal should be interpreted carefully, particularly from coniferous forests where emissions of 2-Methyl-3-buten-2-ol, which also determined at mass 69, take place.

## 2.3 Other components and meteorology

Concentrations of nonmethane hydrocarbons (NMHCs) were measured with APHA-360 instrument (Horiba Company, Japan). This gas analyzer separates $CH_4$ and NMHC by using selective catalytic absorbers and measures the gas concentrations with a flame-ionization detector. The total error in the $CH_4$ and NMHC concentration measurements doesn't exceed ±5 ppb. To supply the flame ionization detector with hydrogen, which is necessary for its operation, the

instrumentation set has hydrogen generators of different types.





CO concentration was measured with TE48S instrument (Thermo Electron Corp). This instrument uses an advanced method based on the measurements with the correlation of gas filters. It allows for measuring background CO concentrations at a level of less than 100 ppb with the total error of ±10 ppb.

$SO_2$ concentration was measured with APSA-360 instrument (Horiba Company, Japan). This instrument measures $SO_2$chemiluminescence induced by UV-radiation. For scheduled calibrations zero-air generator is used.

NO and $NO_2$ concentrations were measured at different times with TE42C-TL instrument (Thermo Electron Corp., USA) and with M200AU instrument produced by Teledyne Corp. (USA). These instruments apply the chemiluminescent method. The minimum NO and $NO_2$ concentrations detectable with these instruments are equal to 0.05 ppb; this allows for measuring the so-called background concentrations not influenced by the pollution sources. $NO_x$ concentration is the sum of NO and $NO_2$ concentrations.

Ozone concentration was measured with Dasibi 1008RS and 1008AH gas analyzers. These instruments are based on the photometric method. They allow measuring the ozone concentration in the range from 1 to 1000 ppb with a total error of ±1 ppb. These instruments undergo scheduled calibrations against the secondary standard, the O3-41M No. 1294 instrument, which undergoes in its turn annual calibrations against the SRP No. 38 standard owned by the Mendeleev Research Metrology Institute (Russia).

For monitoring of meteorological parameters, the instrumentation set was constructed including the following instruments: acoustic anemometer ACAT-3M (Russia), Skaneks (Russia), Driesen&Kem (Germany), and Vaisala (Finland). The temperature profile up to a height of 600 m was measured with MTP-5 meteorological temperature profiler (ATTEX Company, Russia).

## 2.4 Data processing

The major problem of the TROICA data analyses is a correct elimination of screening effects (relative to regional scale pollution sources) produced by local pollution sources along the railroad. Except for small areas of biomass burning and smoldering in the vicinity of railway, such sources are mainly of anthropogenic origin and characterized by highly limited spatial extents (and, hence, an impact upon a chemical composition), so they can be effectively filtered out by applying some objective criteria to the original 10-second dataset. Commonly, the chemical impact is characterized by strong short-time variations in CO, $O_3$, and $NO_x$ abundance as well as increase in $NO/NO_2$ ratio well above its regional characteristic value of ~0.1. Upon a set of trials, we found that eliminating data points satisfying the criteria $NO/NO_2 > 0.2$ is sufficient to obtain a robust statistics on measurement data concerning the regional effects of anthropogenic pollution. According to Fig. 2, such an approach results in eliminating ~25% data from the subsequent analyses with the remaining data (we call it "filtered dataset" henceforth) being safely attributed to the local to regional-scale effects produced by sustained anthropogenic sources.



Statistical and graphical data analysis was performed with a free software environment for statistical computing and graphics, R (http://www.r-project.org/), and with the data analysis and graphing software, OriginPro (OriginLab Corporation).

Statistical summary of the filtered dataset is shown in Table 1. In the east segment, toluene and benzene concentrations and their variations are somewhat higher compared to those for the west segment. As it was discussed above, this feature is most probably attributed to the observed differences in meteorological conditions, as the regional anthropogenic emission sources are not expected to vary at any appreciable rate on sub-seasonal scale. Although atmospheric dilution rates and advection paths (see Figure 1) do vary significantly between the east and west segments, spatial localization of elevated rates in toluene and benzene near their emission sources (mostly large towns and their vicinities) due to their relatively short chemical lifetime strongly restrict the impact of meteorology on their near-surface abundance, at least at regional scale. Consequently, the observed systematic differences in toluene and benzene statistics between east and west segments are relatively small for both mean and percentile values.

## 3 Results and discussion

### 3.1 Spatial distribution of benzene and toluene along the Trans-Siberian railway

The areas adjacent to the Trans-Siberian railway are markedly different in amount of urbanization and anthropogenic load. Eighty-seven towns located immediately on the railway, sixty-eight towns are in the Ural mountain region and West Siberia with the remaining ones located in the East Siberia and the Far East. Yet, it is in the area of first tens to hundreds of kilometers from the Trans-Siberian railway where the most significant regional anthropogenic sources are commonly located in all the regions considered. Fig. 3 shows 10-min averages of the surface concentrations of benzene and toluene derived from the filtered 10-second dataset for the east and west segments of the TROICA-12 route. Simultaneous measurements of the surface NMHCs, CO, and $SO_2$ concentrations are also shown in the figure to specify primary sources of benzene and toluene (see more discussion in p. 3.3).

The observed peaks in volume concentrations shown in the figure are spatially connected to the most significant regional anthropogenic sources along the railway (mostly in urban environment), the latter are expected to be mainly motor vehicle transport, industry, and central heating, and power plants.

Although exact quantitative input of various types of toluene and benzene sources into the observed concentrations remains unknown, some qualitative assessment of their relative importance can be done from a ratio of toluene to benzene (T/B). The T/B ratio is frequently used as an indicator of motor transport emissions, since toluene content in the gasoline and automobile exhausts is in 3–4 times higher than the benzene one (D. Brocco et al., 1997). Therefore, T/B ≈ 1–3 is widely accepted to indicate motor vehicle transport, whereas T/B>3 points out to industrial exhausts (Tiwari et al., 2010; Shaw et al., 2015; Carballo-Pat, 2014). Furthermore, the chemical activity of toluene in the atmosphere is approximately 5 times



higher than that of benzene. Hence, the T/B ratio can serve as indicator of chemical aging and, consequently, proximity of the sampled air to the associated pollution source (Mugica et al., 2003; Tiwari et al., 2010; Carballo-Pat, 2014; Shaw et al., 2015).

Following our previous study on greenhouse gases emissions (Berezina et al., 2014), in present analysis, we divide the whole path from Moscow to Vladivostok into 6 lengthy segments according to climatological conditions and anthropogenic load intensity: European Russia (ER), Ural mountain region (UR), southern parts of West (SWS), Central (SCS), and East (SES) Siberia, and Far East (FE) (see Fig.1). Statistics for benzene, toluene, and T/B from 10-second filtered data for different regions are shown in Table 2. The highest concentrations of benzene and toluene are observed in ER, UR, and SCS, which are the regions of most significant anthropogenic emission sources and proximity of the TROICA route to the strong pollution sources. Since low/high T/B ratio measured at a given location can be equally attributed (in the absence of a prior information) either to a photochemically aged/young air mass or specific chemical composition of the primary pollutants affected the measured air mass, we must distinguish between the both factors to use T/B ratio as characteristic of the associated emission source. The problem is farther complicated by that the measured air represents commonly a mixture of air parcels with different photochemical age and/or anthropogenic loading. One partial solution consists in separating air masses according to their chemical aging (transport times from the regional pollution sources) based on some additional data on their chemical composition and/or transport times.

In present work we utilize the former approach by invoking contemporary measurements of NO and $NO_2$ to distinguish between clean remote air ($NO_x < 2$ ppb), moderately polluted air ($2 \leq NO_x < 20$ ppb) and highly polluted air ($NO_x \geq 20$ ppb), the letter being representative for urban and suburban environment. Here, the exact threshold values of $NO_x$ were chosen basing on our experience in processing multiple data sets from TROICA campaigns as well as continuous measurements of $NO_x$ at ZOTTO Tall Tower – remote cite in Central Siberia, which is occasionally affected by transport of polluted air from major regional anthropogenic sources in the south Siberia (Vasileva et al., 2011). The statistical analysis of the filtered dataset showed that about 80% of the data accounts for moderately polluted air, 18% - for clean remote air and only about 2% - for highly polluted air. Thus, the measurements in TROICA-12 campaign were performed mostly in moderately polluted urban atmosphere ($2 \leq NO_x < 20$ ppb), with maximum values of toluene and benzene reaching 45.6 ppb and 36.5 ppb, correspondingly. These values are significantly less than their one-time maximum allowable concentrations (94 and 159 ppb for benzene and toluene, correspondingly).

Since high ($\geq 2$ ppb) NOx are found to be robust characteristic of photochemically young air (Vasileva et al., 2011), we use associated T/B ($NO_x > 2$ ppb) values to infer some qualitative information on the pollution sources characteristic for the given region. According to Table 3, average and median values of T/B ($NO_x > 2$ ppb) are in the ranges 1.8–2.6 and 1.1–1.6, respectively, which is well below commonly accepted threshold value for the motor vehicle exhausts (T/B<3). Yet, significantly higher T/B ratios (the column $P_{90}$ of the table) are found in all the regions suggesting the other important regional sources of air contamination. These high values were measured commonly as short-lasting events where mobile laboratory crossed the plumes of strongly contaminated air originated from large upwind emission sources, as evidenced





from air trajectory analyses and a prior data on industrial facilities. In such cases, high toluene and benzene concentrations are accompanied with enhanced levels of NMHC, CO, and $SO_2$ as well, indicating the petrochemical and refining exhausts. These include heat and power plants in the vicinities of large towns: Perm, Tyumen, Omsk, Ulan-Ude, Chita, Khabarovsk (Fig.3a), and Ekaterinburg, Tyumen, Krasnoyarsk, Kansk, Irkutsk, Mogocha, Birobidzhan, Khabarovsk (Fig.3b).

Assuming that high (>2 ppb) $NO_x$ is a signature of freshly contaminated air, we estimated the bulk contribution of motor vehicle exhausts, $\chi(\%)$, to the near-surface abundance of  toluene and benzene along the Trans-Siberian railway from a simple relation:

$$\chi(\mathrm{T}) = \frac{\overline{T}_{low} \cdot t_{low}}{\overline{T}_{low} \cdot t_{low} + \overline{T}_{high} \cdot t_{high}} \cdot 100\% \ ,$$

(1)

with similar relation for benzene, where $t_{low}$, $t_{high}$ are fraction of measurement time within air masses having low (< 3 ppb) and high (≥3 ppb) T/B ratios and $NO_x$>2 ppb ($t_{low} + t_{high} = 1$), and $T_{low}$, $T_{high}$ are average toluene concentrations in the low- and high-T/B air masses, correspondingly (see Table 4).

One can see from Table 4, that motor vehicle exhausts accounts for ~ 90% of benzene levels and ~ 65% of toluene levels during the campaign. The major conclusion is that it is the motor vehicle exhausts, which are the most significant

anthropogenic source of air pollution by toluene, and benzene in all the areas adjacent to the TROICA-12 route (densely populated areas along the Trans-Siberian Railroad).

One can see from Table 4, that motor vehicle exhausts accounts for ~ 90% of benzene levels and ~ 65% of toluene levels during the campaign, that is, benzene emission from motor vehicle exhausts is 25% lower than the toluene one. It is comparable with the relative source contributions for benzene, and toluene presented in (Karl et al., 2009). Thus, the motor

vehicle exhausts are the most significant anthropogenic source of air pollution by toluene, and benzene in all the areas adjacent to the TROICA-12 route (densely populated areas along the Trans-Siberian Railroad).

Benzene levels from the TROICA-12 campaign are broadly comparable in magnitude with other published data on their abundance in the summer urban and rural continental surface air (Elansky et al., 2000; Barletta et al., 2005; Na et al., 2005;Parra et al., 2006; Hoque et al., 2008; Tiwary et al., 2010; Seco et al., 2013; Wagner et al., 2014). However, toluene

levels are often less than the published ones. It is possibly due to the more significant contribution of evaporative and industrial emissions in toluene levels (Karl et al., 2009) than this of mobile transport exhausts (which are about 65% for toluene from TROICA-12 measurements).

### 3.2 Diurnal variations of benzene and toluene

To determine the contribution of diurnal variations of benzene and toluene to their surface levels and spatial variability, we

analyzed their hourly mean concentrations measured in the campaign.





Substantial amount of observations at site (Filella et al., 2006, Zalel et al., 2008, Tiwary et al., 2010; Wagner et al., 2014) reported on the highest concentrations of benzene and toluene in the morning and evening hours due to increase in the motor vehicle transport exhausts at this time. Contrary, no clear diurnal variation of benzene and toluene was observed in the TROICA-12 campaign (Fig.4). We suppose that it is due to spatially smoothing of their diurnal variations in the absence of

high pollution, which is clearly seen from median values. Somewhat higher levels in the morning (at 4-5 a.m.) and in the evening (at 8 and 11 p.m.) are most probably due to accumulation of benzene and toluene in the stable atmospheric conditions in the vicinities of their regional sources transected on the TROICA route. Episodic crossing of anthropogenic pollution plumes during the campaign caused the midday peak (which falls occasionally at ~12 a.m.), seen on the Figure 4 from $P_{90}$ profile.

Thus, diurnal variations of planetary boundary layer mixing regime does not contribute significantly to the spatial variability of benzene and toluene along the Trans-Siberian railway in the TROICA-12 campaign.

### 3.3 Benzene and toluene levels in urban and rural surface air

The surface concentrations of VOCs, $NO_x$, CO, and $SO_2$ are commonly found to be notably higher in urban areas (as it would be expected), as most of regional pollution sources are located in cities and their suburbs (Table 5). It was found,

however, that the highest (>P95) concentrations of all the pollutants including benzene and toluene were measured outside the cities so they can not be attributed to direct impact of urban pollution sources. A closer examination shows that these events are most probably connected to specific transport conditions favorable for maintaining anthropogenic plumes from large upwind sources, i.e. towns and industrial manufactures away from the railroad, as highly coherent structures at time scales of the order of few to ten hours. The most prominent events of crossing industrial plumes took place in ETR (up to 37

20  ppb for benzene), SCS (up to 39 ppb for benzene), SES (up to 46 ppb for toluene) and FE (up to 41 ppb for benzene).

One can see from Table 6 that there is a meaningful correlation (R≈0.5) between benzene and toluene both in urban and rural areas. In urban environments, there are also high correlations between benzene and toluene and CO (R≈0.6), as well as between $NO_x$ and CO, which points out to the motor vehicle transport as their common emission source. Benzene is also found to appreciably correlate with NMHC which can indicate partial input from the industrial exhausts: hydrocarbon

processing, refining industry, fuel transportation and storage, tanks and pipeline leaks, etc.

In rural areas (identified as those outside the towns with additional constraint NOx<0.2 ppb, see p.3.1), correlation between all the species studied is very poor except for that between benzene and toluene. Evidently, diversity of transport pathways, photochemical aging, as well as irreversible mixing of air masses subjected to different rates of anthropogenic contamination precludes direct quantification of primary pollutant sources for the substantial part of the TROICA route away from areas of

their immediate impact.

To study atmospheric pollution in Russian cities along the Trans-Siberian railway, 29 cities were selected for which the total amount of measurement time was at least 25% of the whole residence time in the city. The highest concentrations of benzene



and toluene (up to 5 ppb) are observed in the industrial towns: Perm, Kirov (European Russia); Kungur and Yekaterinburg (south Ural mountain region), Tyumen (West Siberia), Angarsk, Irkutsk, and Ulan-Ude (East Siberia), Birobidzhan, Khabarovsk (Far East region) (Fig. 5). In these cities the highest levels of NMHC, NOx, and CO were also measured (Fig.6). Evidently, high CO abundances found in some of these towns point out to the significant contribution to the overall pollution

rates from refineries and central heating and power plans, which is also confirmed by high T/B ratio (>3 − 4 basing on P90 regional values). The specific T/B ratios also show that Khabarovsk, Birobidzhan, Skovorodino, Tulun, Tajshet, and Tyumen are mainly polluted by industrial emissions, whereas Vladimir, Kungur, Yurga, and Krasnoyarsk − by transport exhausts. In other cities, motor vehicle transport is found to be a main pollution source although the contribution from industrial sources is also important as seen from significantly higher P90 values comparing to the average one calculated for rural regions (see

Fig.5). Benzene and toluene surface levels in medium-sized towns of Siberia (Achinsk, Taishet, Nizhneudinsk, et al.) are close to an average rural concentration calculated for the campaign (less than 0.3 ppb).

Unfortunately, exact quantification of inputs from various types of sources into anthropogenic contamination of urban air is inhibited when using TROICA data due to very limited amount of observation collected within a particular town. Considering that transport emissions occur when the T/B is in the range from 1 − 3 (Mugica et al., 2003; Tiwari et al.,

2010; Carballo-Pat, 2014; Shaw et al., 2015), and supposing well mixed conditions such that each measured air parcel represent a uniform mixture of pollutants from various sources within a town, we found that motor vehicle transport accounts for approximately 75% of anthropogenic emissions in the Russian cities along the Trans-Siberian railway with the remaining 25% are attributed to industrial sources (Fig.7). These estimates correspond well to those derived in section 3.1 basing on the whole TROICA dataset. As seen from Figure 8, T/B ratio for vehicle urban exhausts in the Russian cities along

the Trans-Siberian railway is usually in the range of 2.3–2.8.

### 3.4 Contribution of VOCs to ozone formation potential over Russia

Along the whole route of the TROICA campaign the lower troposphere chemical regime is found to be essentially a NOx sensitive, both in rural and urban environments, with typical morning NMHC/NOx ratios being well above 20. Hence, ozone production rates are expected to be controlled by regional NOx emissions (Silman, 1999) and their complex interplay with

both natural and anthropogenic sources of VOCs. As it was mentioned above, the meteorological conditions during the most of the TROICA campaign were favorable for both studying chemical composition of fresh air masses contaminated by regional sources as well as for ozone production from the emitted precursors due to high daytime surface air temperatures and solar radiation. To estimate the impact of the measured VOCs on regional ozone production, we employ the widely used quantities: propylene-equivalent concentration (PE) and ozone-forming potential (OFP) (Carter, 1994; So and Wang, 2004),

which utilize the measured concentrations of VOCs along with their reactivity with hydroxyl radical. These coefficients are defined as:

$$PE_{VOC} \quad [ppbC] = C_{VOC} \times k_{OH,VOC} / k_{OH,propylene} , \qquad (2)$$



$$\mathrm{OFP_{VOC}[\mu g/m^3]} = C_{\mathrm{VOC}} \times MIR_{VOC}, \tag{3}$$

where $C_{\mathrm{VOC}}$ is a VOC concentration having dimension of ppbC and μg/m$^3$ in (2) and (3), correspondingly, $k_{\mathrm{OH,VOC}}$ is the rate constant for the reaction of VOC with OH-radical, $k_{\mathrm{OH,propylene}}$ is the rate constant for the reaction between OH and propylene, and $MIR_{VOC}$ is a maximum incremental reactivity. The latter is a dimensionless quantity defined as gram of O$_3$

produced per gram of the VOC, which equals to the maximum ozone concentration formed from chemical destruction of the given VOC.

We calculate PE and OFP values for benzene and toluene basing on the daytime observations from 12 a.m. − 5 p.m., the time for which the highest correlations between ozone and its precursor species were observed in TROICA as well. The calculated PE and OFP values were compared against those for isoprene, the latter being known as one the most important

biogenic ozone precursors in rural as well as urban settings (Chameides et al., 1988; Fuentes et al., 2000; Wagner et al., 2014).

As seen from Table 7, the average value of OFP of isoprene along the TROICA route is much higher compared to those for benzene and toluene, owing to relatively high near-surface abundance of the former (approx. 3 times as much as that for the sum of benzene and toluene) and exceptionally high reactivity with hydroxyl radical. According to Fig.8, the process of

oxidizing of isoprene proves to be the most important chemical source of ozone in all the regions along the TROCA route, as it could be expected if one takes into account that the most part of the railway crosses the areas with very weak to moderate anthropogenic load. The highest OFPs of isoprene seen on the figure in the Far East are due to its high biogenic emissions from broad-leaved forests as well as high surface air temperatures measured in this region in the both east and west segments of the route.

As seen from figure 8, the OFPs of benzene and toluene do not have significant large scale spatial variations along the railway, contrary to that for isoprene. The highest OFPs shown as peaks on figure 8 are spatially connected to large towns and their vicinities along the railway where the relative input of benzene and toluene into ozone production reaches as high as 60–70% compared to that of isoprene. This supports our general notion of the reduced impact of regional anthropogenic sources on the regional ozone budget compared to long range advection and regional biogenic VOC emissions (Shtabkin et

al., 2016). Taking into account the important role of biogenic emissions of isoprene in the regional ozone photochemistry, the detailed analyses of isoprene observations from the TROICA campaigns should be done in a separate publication.

## 4 Summary

Surface concentrations of important anthropogenic VOCs, benzene and toluene, as well as inorganic compounds were measured simultaneously along the Trans-Siberian railway on a mobile railway laboratory in the TROICA-12 campaign in

summer 2008. It is demonstrated that the TROICA-12 measurements were carried out mostly in moderately polluted ($2 \leqslant$ NOx<20 ppb) environment (~78% of measurements) with the remaining part of measurement time divided between weakly



polluted (NOx ⩽ 2 ppb) and highly polluted (NOx>20 ppb) urban environment (20 and 2% of measurements, correspondingly). Maximum values of benzene and toluene during the campaign reached 36.5 ppb and 45.6 ppb, correspondingly, which is significantly less than their one-time maximum permissible concentrations (94 and 159 ppb for benzene and toluene, correspondingly). Although the weather conditions during the major part of the TROICA campaign

were favorable for accumulating anthropogenic pollutants in the lower atmosphere, the absence of clear diurnal variations of benzene and toluene along with their low abundancies apart from the immediate vicinity of large towns and industrial manufactures points out to mostly unpolluted air conditions along the Trans-Siberian railway during the campaign.

We estimated that motor vehicle exhausts accounts for ~ 90% of benzene levels and ~ 65% of toluene levels during the campaign, with the remaining 10% and 25%, correspondingly, provided by other important regions anthropogenic sources:

industrial enterprises, transportation and storage of VOCs etc.

The highest near-surface abundances of benzene and toluene, both in urban environment and on the regional scale, are observed in areas with highest anthropogenic loading. These are industrial regions of the South Ural, European Russia, and south of Central Siberia, where spatially averaged concentrations of benzene and toluene, representative for rural conditions, equal to ~0.3 and ~0.4 ppb, correspondingly. The vehicle emissions in these regions consist the major part of total

anthropogenic pollution, with typical ration of T/B be 2.2 – 2.3. Similarly, the highest concentrations of benzene (up to 5 ppb) and toluene (up to 7 ppb) along with high levels of NMHC, CO and $NO_x$ are observed in the following industrial towns: Perm, Kirov (European Russia); Kungur and Yekaterinburg (south Ural mountain region), Tyumen (West Siberia), Angarsk, Irkutsk, and Ulan-Ude (East Siberia), Birobidzhan, Khabarovsk (Far East region).

Considering that transport emissions occur when the T/B is in the range from 1 – 3 and supposing well mixed conditions

such that each measured air parcel represent a uniform mixture of pollutants from various sources within a town, we found that motor vehicle transport accounts for approximately 75% of anthropogenic emissions in the Russian cities along the Trans-Siberian railway with the remaining 25% are attributed to industrial sources. T/B ratio for vehicle urban exhausts in the Russian cities along the Trans-Siberian railway is usually in the range of 2.3–2.8.

The contribution of benzene and toluene to the local photochemical ozone production along the Trans-Siberian railway is

generally not significant compared to biogenic VOCs in rural environment and reaches as much as 16% of that of isoprene. However, in large towns the contribution of benzene and toluene to ozone formation reaches 60–75% compared to isoprene, supporting important role of anthropogenic sources in local pollution.

*Acknowledgements*. The authors thank Shumsky R.A. for an active participation in designing of the measurement set of the

mobile laboratory and controlling of its correct work and Lavrova O.V. for an active participation in the campaign and careful diary observations. This study was supported by the Russian Science Foundation (grant no. 14-47-00049), by the Russian Foundation for Basic Research (grant no. 15-35-50970) and contributes to the Pan-Eurasian Experiment (PEEX) Program research agenda.





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





**Table 1: Statistics for the filtered original 10-sec data of benzene and toluene (in ppb) from the TROICA-12 campaign (E – East segment; W – West segment). N – total number of the original 10-sec filtered samples; σ – standard deviation, P – percentiles.**

| TROICA segment | N | Mean | σ | Min | Max | P10 | P50 | P90 |
|---|---|---|---|---|---|---|---|---|
| **Benzene** | | | | | | | | |
| **E** | 39873 | 0.23 | 0.60 | 0.01 | 36.51 | 0.05 | 0.15 | 0.40 |
| **W** | 36322 | 0.18 | 0.27 | 0.01 | 14.32 | 0.05 | 0.13 | 0.35 |
| **Toluene** | | | | | | | | |
| **E** | 39873 | 0.34 | 0.79 | 0.01 | 45.58 | 0.07 | 0.21 | 0.62 |
| **W** | 36322 | 0.27 | 0.48 | 0.01 | 25.89 | 0.06 | 0.17 | 0.50 |





**Table 2: The regional statistics of benzene, toluene (in ppb) and T/B along the Trans-Siberian railway: ER - European Russia, UR - Ural mountain region, SWS - southern parts of West Siberia, SCS - southern parts of Central Siberia, SES - southern parts of East Siberia, FE - Far East.**

| Region | N | Mean | σ | Min | Max | $P_{10}$ | $P_{50}$ | $P_{90}$ |
|--------|------|------|------|------|------|------|------|------|
| Benzene | | | | | | | | |
| ER | 7456 | 0.28 | 1.10 | 0.01 | 36.51 | 0.08 | 0.16 | 0.43 |
| UR | 5464 | 0.28 | 0.42 | 0.01 | 12.22 | 0.08 | 0.20 | 0.49 |
| SWS | 10714 | 0.22 | 0.44 | 0.01 | 22.28 | 0.07 | 0.14 | 0.37 |
| SCS | 13120 | 0.25 | 0.64 | 0.01 | 35.35 | 0.07 | 0.16 | 0.45 |
| SES | 26228 | 0.20 | 0.26 | 0.01 | 14.99 | 0.07 | 0.15 | 0.36 |
| FE | 13212 | 0.22 | 0.28 | 0.01 | 15.81 | 0.08 | 0.17 | 0.38 |
| Toluene | | | | | | | | |
| ER | 7456 | 0.35 | 0.65 | 0.01 | 32.10 | 0.09 | 0.23 | 0.67 |
| UR | 5464 | 0.39 | 0.86 | 0.01 | 32.82 | 0.10 | 0.27 | 0.70 |
| SWS | 10714 | 0.28 | 0.56 | 0.01 | 17.26 | 0.07 | 0.17 | 0.48 |
| SCS | 13120 | 0.37 | 0.89 | 0.01 | 38.58 | 0.08 | 0.21 | 0.71 |
| SES | 26228 | 0.31 | 0.59 | 0.01 | 45.58 | 0.09 | 0.21 | 0.55 |
| FE | 13212 | 0.28 | 0.63 | 0.01 | 40.67 | 0.08 | 0.18 | 0.45 |



**Table 3: T/B ratio for moderately to strongly polluted air (NOx > 2 ppb).**

| Region | N | Mean | σ | Min | Max | $P_{10}$ | $P_{50}$ | $P_{90}$ |
|--------|------|------|------|------|--------|------|------|------|
| ER | 4247 | 2.20 | 2.63 | 0.04 | 41.71 | 0.49 | 1.53 | 4.26 |
| UR | 3801 | 2.26 | 2.90 | 0.04 | 49.00 | 0.52 | 1.51 | 4.40 |
| SWS | 6685 | 2.27 | 3.28 | 0.01 | 57.65 | 0.40 | 1.37 | 4.63 |
| SCS | 10732 | 2.28 | 3.08 | 0.02 | 72.13 | 0.48 | 1.50 | 4.44 |
| SES | 21824 | 2.58 | 3.73 | 0.03 | 187.77 | 0.49 | 1.61 | 5.30 |
| FE | 11645 | 1.84 | 3.01 | 0.01 | 87.86 | 0.33 | 1.10 | 3.62 |





**Table 4: The bulk contribution of motor vehicle exhausts, $\chi\,(\%)$, to the near-surface abundance of toluene and benzene along the Trans-Siberian Railroad (see Eq. 1).**

| TROICA Segment | $\overline{T}_{low}$ | $t_{low}$ | $\overline{T}_{high}$ | $t_{high}$ | $\chi\,(\%)$ |
|---|---|---|---|---|---|
| | | | Benzene | | |
| East | 0.274 | 0.77 | 0.123 | 0.22 | 89 |
| West | 0.274 | 0.80 | 0.119 | 0.20 | 90 |
| | | | Toluene | | |
| East | 0.316 | 0.77 | 0.603 | 0.22 | 65 |
| West | 0.333 | 0.80 | 0.704 | 0.20 | 65 |





**Table 5: Surface levels of the studied impurities in urban and rural areas along the Trans-Siberian railway. All impurities in ppb except for NMHC and CO (ppm).**

| Compound | N | Mean | σ | Min | Max | P10 | P50 | P90 |
|---|---|---|---|---|---|---|---|---|
| **Urban** | | | | | | | | |
| $C_6H_6$ | 10492 | 0.37 | 0.33 | 0.01 | 2.94 | 0.12 | 0.26 | 0.67 |
| $C_5H_8$ | 10492 | 0.70 | 1.34 | 0.04 | 22.56 | 0.17 | 0.39 | 1.30 |
| NMHC | 7571 | 0.26 | 0.24 | 0.09 | 2.65 | 0.14 | 0.20 | 0.39 |
| $NO_x$ | 11052 | 11.95 | 16.14 | 0.75 | 205.64 | 2.78 | 7.57 | 23.82 |
| CO | 7239 | 0.27 | 0.08 | 0.14 | 0.68 | 0.20 | 0.25 | 0.36 |
| $SO_2$ | 7518 | 1.38 | 1.16 | 0.04 | 10.74 | 0.35 | 1.16 | 2.48 |
| **Rural** | | | | | | | | |
| $C_6H_6$ | 65703 | 0.21 | 0.43 | 0.01 | 36.51 | 0.07 | 0.15 | 0.37 |
| $C_5H_8$ | 65703 | 0.28 | 0.47 | 0.01 | 45.58 | 0.08 | 0.20 | 0.51 |
| NMHC | 51497 | 0.16 | 0.09 | 0.03 | 3.42 | 0.11 | 0.14 | 0.21 |
| $NO_x$ | 79941 | 4.38 | 5.07 | 0.56 | 237.98 | 1.46 | 3.38 | 7.70 |
| CO | 50256 | 0.23 | 0.07 | 0.05 | 2.77 | 0.17 | 0.22 | 0.28 |
| $SO_2$ | 53502 | 1.22 | 0.82 | 0.10 | 9.80 | 0.30 | 1.10 | 2.20 |



**Table 6: Pearson correlation matrix for urban and rural measurements along the Trans-Siberian railway. R ≥ 0.5 are shown in bold. Asterisks show significant correlations for P = 0.05. All impurities in ppb except for NMHC and CO (ppm).**

| | $C_6H_6$ | $C_5H_8$ | $NO_x$ | CO | $SO_2$ | NMHC |
|---|---|---|---|---|---|---|
| **Urban** | | | | | | |
| $C_6H_6$ | 1 | **0.53*** | 0.23* | **0.57*** | 0.35* | **0.47*** |
| $C_5H_8$ | **0.53*** | 1 | 0.15* | **0.49*** | 0.18* | 0.21* |
| $NO_x$ | 0.23* | 0.15* | 1 | **0.47*** | 0.25* | 0.21* |
| CO | **0.57*** | **0.49*** | **0.47*** | 1 | 0.25* | 0.21* |
| $SO_2$ | 0.35* | 0.18* | 0.25* | 0.25* | 1 | 0.01 |
| NMHC | **0.47*** | 0.21* | 0.21* | 0.21* | 0.01 | 1 |
| **Rural** | | | | | | |
| $C_6H_6$ | 1 | **0.52*** | 0.10* | 0.11* | 0.08* | 0.21* |
| $C_5H_8$ | **0.52*** | 1 | 0.17* | 0.14* | 0.07* | 0.17* |
| $NO_x$ | 0.10* | 0.17* | 1 | 0.22* | 0.16* | 0.16* |
| CO | 0.11* | 0.14* | 0.22* | 1 | 0.13* | 0.18* |
| $SO_2$ | 0.08* | 0.07* | 0.16* | 0.13* | 1 | 0.13* |
| NMHC | 0.21* | 0.17* | 0.16* | 0.18* | 0.13* | 1 |





**Table 7: Averaged concentrations and photochemical properties of benzene, toluene and isoprene (± standard deviation) from the TROICA-12 campaign.**

| VOC | $10^{12}*K_{OH}$[a] | MIR[b] | Concentration, ppb | OFP[c] ($\mu g/m^3$) | PE[d] ($\mu g/m^3$) |
|---|---|---|---|---|---|
| Benzene | 1.23 | 0.42 | 0.20±0.33 | 0.29±0.48 | 0.06±0.09 |
| Toluene | 5.96 | 2.70 | 0.28±0.51 | 3.10±5.69 | 0.44±0.81 |
| Isoprene | 101.00 | 9.10 | 0.60±0.55 | 16.65±15.19 | 11.52±10.51 |

[a] Rate constants of VOCs with OH at 298 K ($sm^3$ molecule$^{-1}$ s$^{-1}$) (Atkinson, 1989; Atkinson and Arey, 2003).

[b] Maximum incremental reactivity (g $O_3$/g VOC) (Carter, 1994, 1997).

[c] Propylene-equivalent concentration.

[d] Ozone-forming potential.



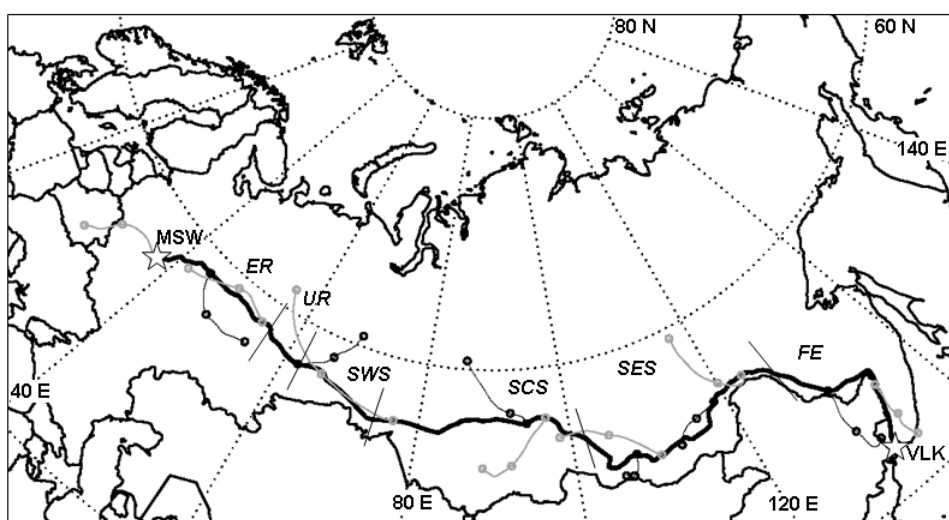

5   **Figure 1: Schematic representation of the TROICA-12 route from Moscow (MSW) to Vladivostok (VLK). Thin solid lines across the route represent approximate boundaries of various geographic regions: European Russia (ER), Ural mountain region (UR), southern parts of West (SWS), Central (SCS), and East (SES) Siberia, and Far East (FE). Back 2-day trajectories with endpoints at Trans-Siberian Railroad at local noon of each successive day of carriage movement are shown for East (black solid) and West (grey solid) routs of the campaign. Open circles mark air particles positions at 0, 24, and 48 hours along the each trajectory.**





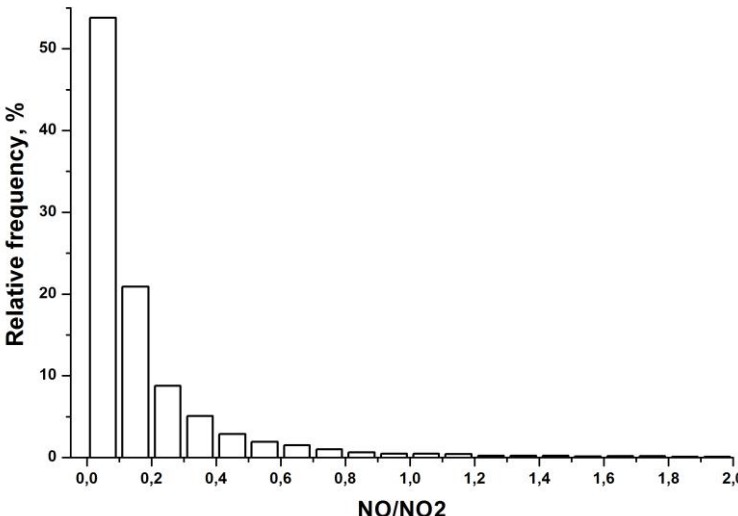

**Figure 2: Relative frequency of NO/NO2 values in the TROICA-12 campaign.**



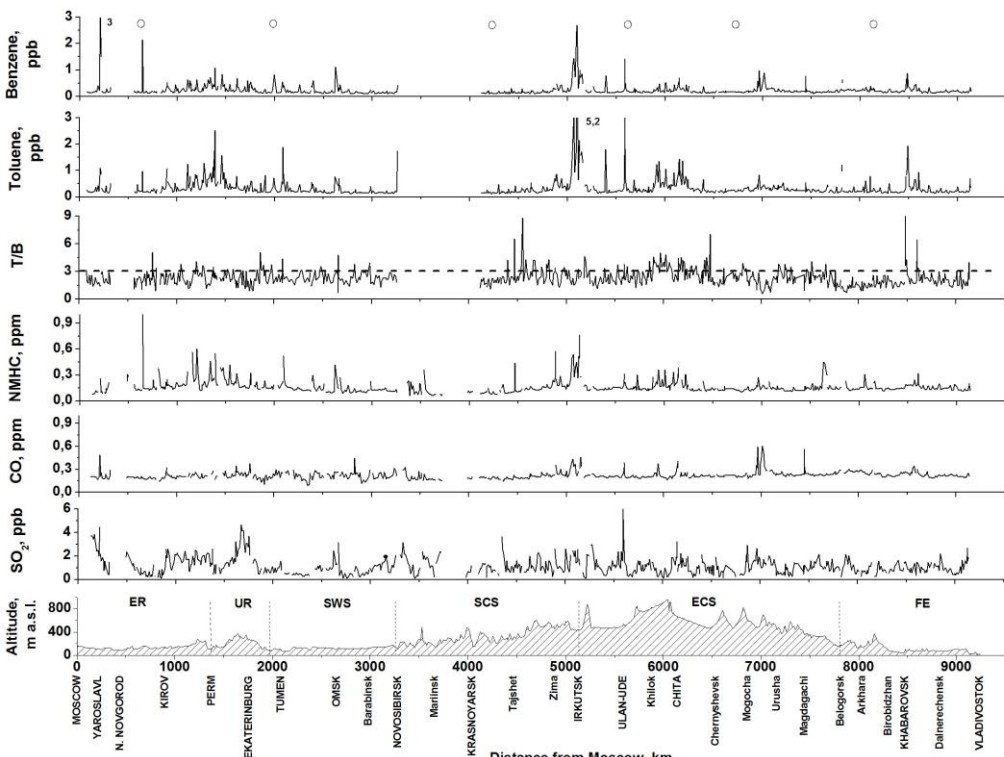

**Figure 3a: 10-minute concentrations of benzene, toluene, T/B ratio as well as NMHC, CO and SO₂ along the Trans-Siberian**
**railway in the East segment of the TROICA-12 campaign. White circles on the top of the figure – the times of local noon. The cities**
**with the population density from 250000 to 1 million and more are shown by large font. A dashed line shows T/B boundary**
**between transport and industrial emissions (Tiwary et al., 2010; Carballo-Pat, 2014).**





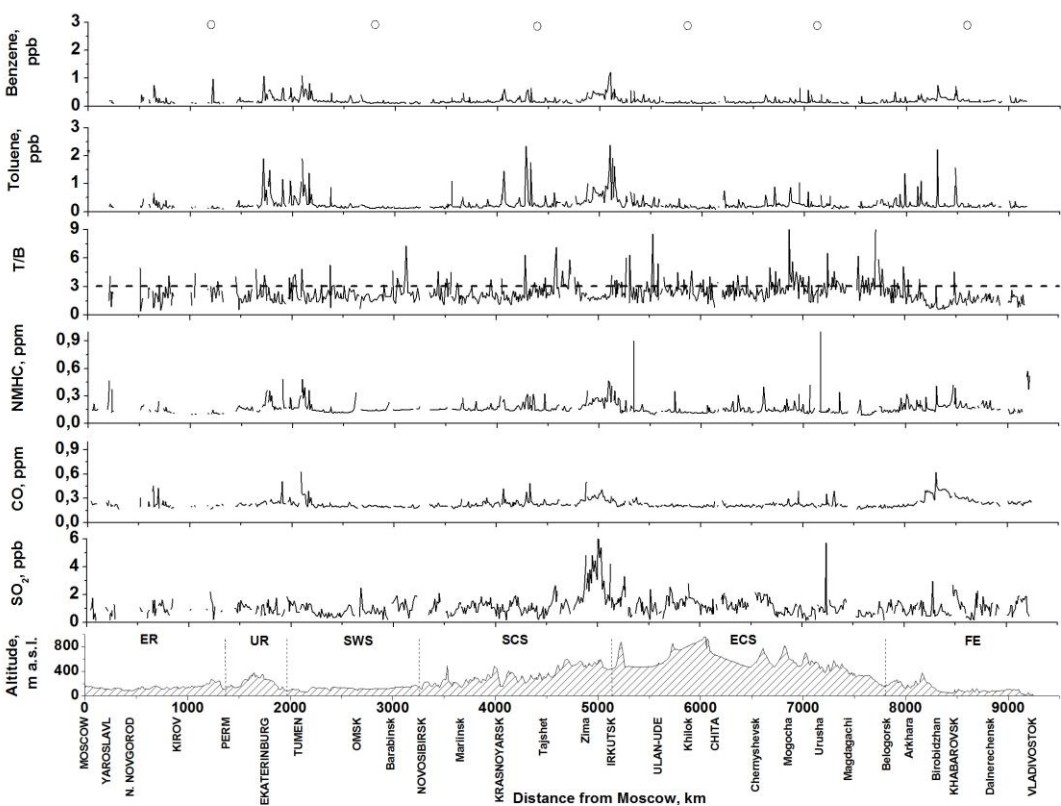

5  **Figure 3b: 10-minute concentrations of benzene, toluene, T/B ratio as well as NMHC, CO and SO₂ along the Trans-Siberian railway in the West segment of the TROICA-12 campaign. White circles on the top of the figure – the times of local noon. The cities with the population density from 250000 to 1 million and more are shown by large font. A dashed line shows T/B boundary between transport and industrial emissions (Tiwary et al., 2010; Carballo-Pat, 2014).**





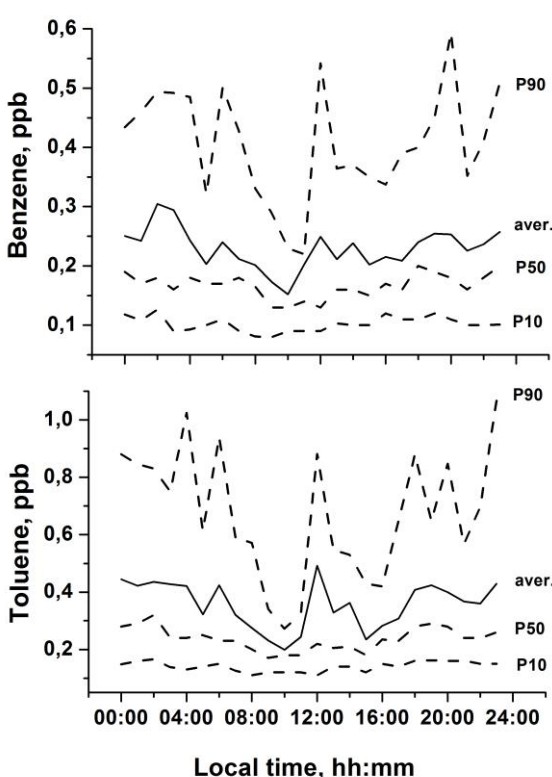

**Figure 4: Diurnal profiles of the surface levels of benzene and toluene from TROICA-12 (hourly mean values): average (solid) and percentiles (dashed).**





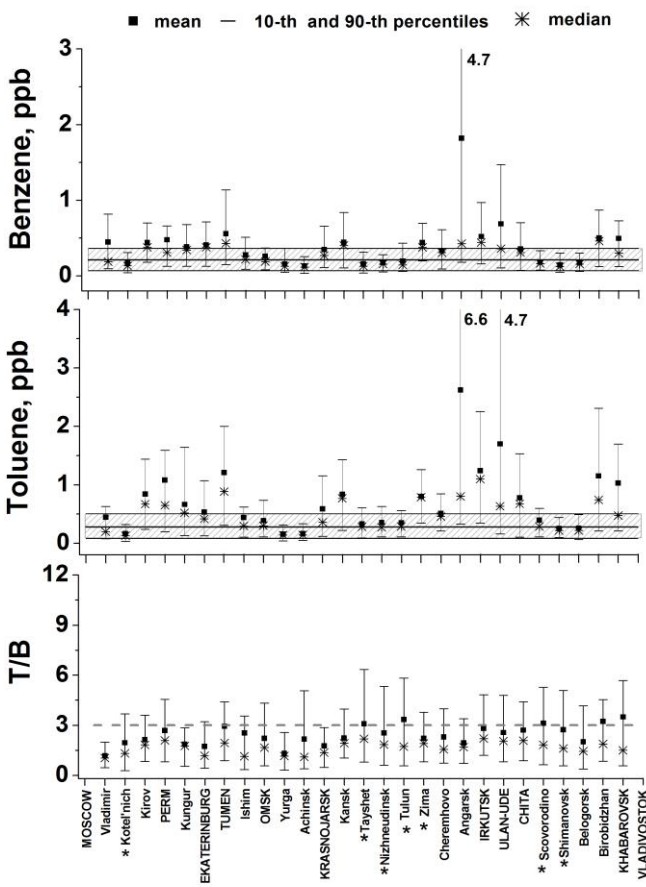

**Figure 5: Benzene, toluene and T/B in Russian cities along the Trans-Siberian railway. The cities with the population density from 250 000 to 1 million and more are shown by large font, the cities with the population density from 50 000 to 250 000 – by small font. Asterisks show the cities with the population density less than 50 000. The cities are shown in accordance with their location along the railway. The shaded area – mean, 10-th and 90-th percentiles calculated from the data measured in rural regions. Dotted line – is a border between vehicle and industrial exhausts (Mugica et al., 2003; Tiwari et al., 2010; Carballo-Pat, 2014; Shaw et al., 2014).**




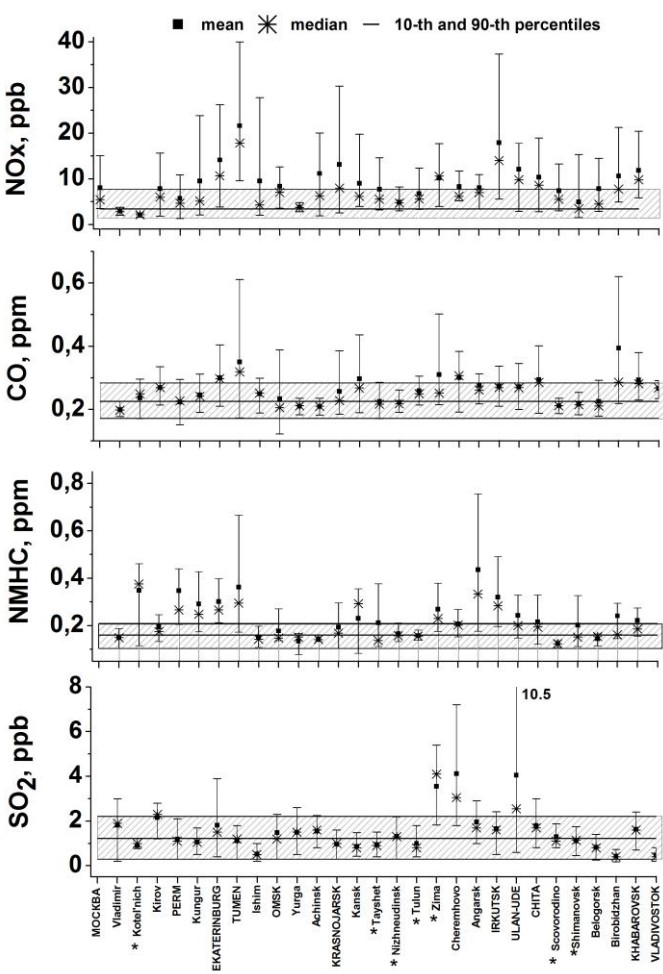

**Figure 6: Inorganic impurities in Russian cities along the Trans-Siberian railway. The cities with the population density from 250 000 to 1 million and more are shown by large font, the cities with the population density from 50 000 to 250 000 – by small font. Asterisks show the cities with the population density less than 50 000. The cities are shown in accordance with their location along the railway. The shaded area – mean, 10-th and 90-th percentiles calculated from the data measured in rural regions.**





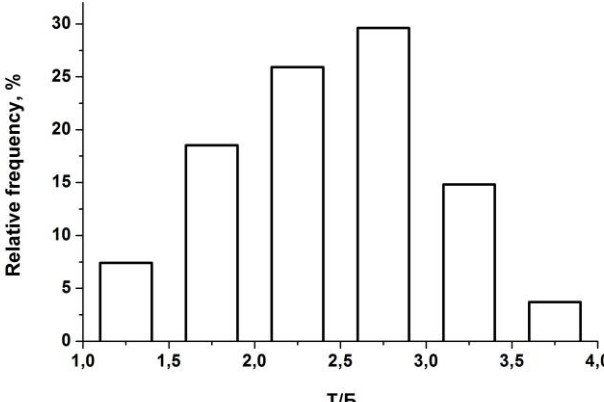

Figure 7: Frequency distribution of T/B ratio for the Russian cities along the Trans-Siberian railway.




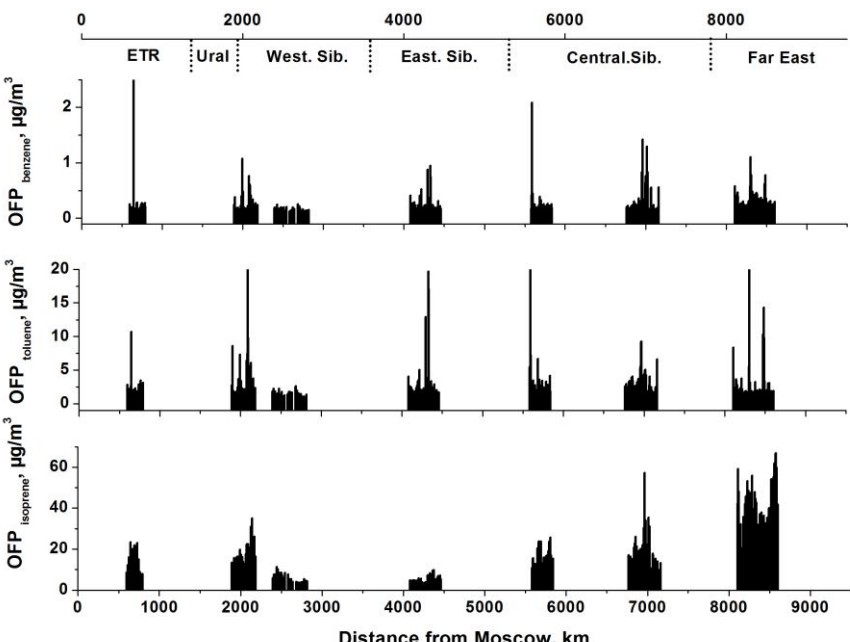

**Figure 8: Ozone formation potential (OFP) along the Trans-Siberian railway from daytime measurements (12 a.m. – 5 p.m. local time).**