# Peer review of "Benzene and Toluene in the surface air of North Eurasia from TROICA-12 campaign along the Trans-Siberian railway"

_Atmospheric Chemistry and Physics, 2016_

## Referee Comment (RC1) · Anonymous Referee #1 · 24 Dec 2016

General The manuscript of "Benzene and Toluene in the surface air of North Eurasia from TROICA-12 campaign along the Trans-Siberian railway" by Skorokhod et al. presented benzene and toluene levels measured by PTR-MS on a mobile laboratory aboard the Trans-Siberian train. In the manuscript the sources of benzene and toluene along the Trans-Siberian railway were investigated mainly using the T/B ratios, and propene-equivalents and OFPs were also discussed. As emission sources of benzene and toluene include not only motor vehicle exhaust and industry emission, but also biomass burning, coal burning and gasoline evaporation, the source attribution need consider contributions from sources other than vehicle exhaust and industry emission. For the comparisons of aromatics' ozone formation relative to isoprene in the rural and

urban areas, it should be noted that aromatic hydrocarbons are quite different from isoprene in reactivity and atmospheric lifetimes, and in their source regions and emission patterns, therefore it is important to figure out new understandings and new findings other than somewhat common senses like larger contributions of benzene and toluene to ozone formations in the urban areas.

Details

1. Page 3, the authors should give more details about where the mobile laboratory was located, in the front of the train or in the end or elsewhere? How to eliminate the interferences from emission inside the train or human activities in the train? 2. Page 4, the calibration method and frequencies for measuring VOCs by PTR-MS and APHA-360 should be stated. 3. Page 8, lines 17 -20 were repeating lines 13-16. 4. Page 8, reference Karl et a., 2009 was not listed in the reference. 5. Page 8, section 3.2, as the benzene and toluene data were measured along the Trans-Siberian railway instead of at a fixed station, the discussions about diurnal variations of benzene and toluene should be careful. It is difficult to say whether emission sources, photochemistry or meteorological conditions had led to the variations. I'd like to suggest deleting this section. 6. Page 9, The correlations coefficients (R) between benzene, toluene, NMHC, CO, NOx, and SO2 were all less than 0.6, that means their R2 were all lower than 0.36. I don't think these can suggest the high or significant correlations between them. Section 3.3 should be rearranged. 7. Table 5 and Table 6, I think the authors wanted to list C7H8 (Toluene) instead of C5H8 (isoprene) as they discussed in section 3.3. 8. Figures 3a, 3b and 8 in the manuscripts are really hard to read. I suggest plotting them in a different way.

---

## Author Comment (AC1) · 26 Jan 2017

The authors thank the anonymous referee #1 for the constructive comments and corrections. They helped us to improve our paper.

1. In the manuscript the sources of benzene and toluene along the Trans-Siberian railway were investigated mainly using the T/B ratios, and propene-equivalents and OFPs were also discussed. As emission sources of benzene and toluene include not only motor vehicle exhaust and industry emission, but also biomass burning, coal burning and gasoline evaporation, the source attribution need consider contributions from sources other than vehicle exhaust and industry emission.

[Figure]

Reply: The referee is certainly right. Biomass burning, coal burning and gasoline evaporation are very important sources of benzene and toluene emissions in the atmosphere. However, taking into account the measurements on a mobile platform, it is difficult to separate the data corresponding to the emissions from different sources (we will be very grateful for more exact recommendations). Therefore, taking into account the fact that vehicle exhaust is one of the main sources of benzene and toluene in anthropogenically polluted atmosphere, we use the well known criteria and the simultaneous measurements of CO, NOx, SO2 and NMHC in the campaign to distinguish the impact of vehicle emissions as well as other local and regional-scale sources (including industrial emissions, coal burning and evaporative emissions) on benzene and toluene levels. This information is also very important for the immense Russian territory. It should be noted, however, that in TROICA-12 experiment no significant biomass burning along the Tran-Siberian railway was observed, so this emission source is not expected to impact significantly on benzene and toluene levels (see corrections at page 9, lines 1-3, and page 12, lines 16-17).

2. For the comparisons of the ozone formation rate from aromatics relative to that due to isoprene in the rural and urban areas, it should be noted that aromatic hydrocarbons are quite different from isoprene in reactivity and atmospheric lifetimes, and in their source regions and emission patterns. Therefore, it is important to figure out new understandings and new findings other than somewhat common senses like larger contributions of benzene and toluene to ozone formations in the urban areas.

Reply: It is well known that aromatic VOCs differ from isoprene by reactivity and sources. However, isoprene is one of the main biogenic VOCs forming ozone in the polluted surface air. So, it was important to reveal a significance of the benzene and toluene impact in ozone formation along the Trans-Siberian railway in contrast to isoprene. These estimations are very important because of a lack of such information for Russian regions. Although some new understandings are certainly necessary.

Details

1. Page 3, the authors should give more details about where the mobile laboratory was located, in the front of the train or in the end or elsewhere? How to eliminate the interferences from emission inside the train or human activities in the train?

Reply: The location of the laboratory is described in the section 2.1 TROICA experiments, page 3, lines 2-3. To reduce a possible influence of emission from human activities in the train, all conveniences were placed at the end part of the train. Thus, their impact is expected to be generally non-significant (Panin et al., 2001). This information was added to the text (page 3, lines 8-12).

2. Page 4, the calibration method and frequencies for measuring VOCs by PTR-MS and APHA- 360 should be stated.

Reply: Agreed. We provide more information about the instruments in the final version of the paper (see the sections 2.2 VOC measurements, page 4, lines 10-20 and 2.3 Other components and meteorology, page 5, lines 8-10).

3. Page 8, lines 17 -20 were repeating lines 13-16 Reply: Agreed. Lines 13-16 are removed from the text.

4. Page 8, reference Karl et al., 2009 was not listed in the reference.

Reply: Agreed. The reference is added to the paper, page 14, line 32.

5. Page 8, section 3.2, as the benzene and toluene data were measured along the Trans-Siberian railway instead of at a fixed station, the discussions about diurnal variations of benzene and toluene should be careful. It is difficult to say whether emission sources, photochemistry or meteorological conditions had led to the variations. I'd like to suggest deleting this section.

Reply: The referee is certainly right about the influence of wide range of factors on the diurnal variations of benzene and toluene obtained from the measurements on a mobile laboratory. However, the statistics calculated for hourly mean values and presented in Fig. 4 allows some differentiating among these factors. Furthermore, the study of

**[ACPD](https://www.atmospheric-chemistry-and-physics.net/)**

Interactive
comment

diurnal variations of benzene and toluene in the campaign is necessary to determine their contribution to spatial variability. The absence of significant diurnal variations of benzene and toluene allows us to neglect them and summarize the data in different spatial scales (for different regions and cities of Russia), as pointed out at page 9 lines 16-18.

6. Page 9, The correlations coefficients (R) between benzene, toluene, NMHC, CO, NOx, and SO2 were all less than 0.6, that means their R2 were all lower than 0.36. I don't think these can suggest the high or significant correlations between them. Section 3.3 should be rearranged.

Reply: As we know, the significance of the correlation coefficients is determined by t-test. According to our statistical calculations, all coefficients presented in Table 6 and shown by asterisks are statistically significant including the relationships between benzene, toluene, NMHC, CO, NOx, and SO2. Some of the coefficients are a little bit less than 0.5. It means that the strength of correlation is moderate (R= 0.3 to 0.5). The correlation coefficients for the relationships between benzene and toluene and CO and benzene in urban areas are > 0.5. The latter may evidence for (potentially) strong relationship (R= 0.5 to 1.0) between these compounds. Thus, the correlation analysis presented in section 3.3 allows us to see a relation between some compounds in urban atmosphere determining by their common pollution sources even in the large spatial scale. Furthermore, we noted that in rural areas correlation between all the species studied is very poor except for that between benzene and toluene. Some terminological corrections are included in the section 3.3 (page 9, lines 28-31).

7. Table 5 and Table 6, I think the authors wanted to list C7H8 (Toluene) instead of C5H8 (isoprene) as they discussed in section 3.3.

Reply: The referee is right. It is corrected.

8. Figures 3a, 3b and 8 in the manuscripts are really hard to read. I suggest plotting them in a different way.

[Figure]

Reply: The experience of previous presentations of a big TROICA dataset showed that it is rather difficult to present the data in a lot better way, however, we tried to improve the quality of the figures.

Please also note the supplement to this comment:
http://www.atmos-chem-phys-discuss.net/acp-2016-858/acp-2016-858-AC1-supplement.pdf
* * *
[Figure]

**a)**

Fig. 1.

**b)**

**Fig. 2.**

[Figure]

Fig. 3.

**Supplement:**

[revised manuscript text omitted]

---

## Referee Comment (RC2) · Anonymous Referee #2 · 7 Feb 2017

This manuscript contains a potentially interesting set of measurement data collected across Siberia. It appears that the measurements were properly conducted, and that the presented results are, at least mostly, scientifically sound. While the paper is relatively well structured, the overall presentation of the results needs to be improved here and there before I can recommend acceptance for publication. My detailed comments in this respect are given below.

Section 3.1

Have the authors considered plotting the concentrations in Figure 3 in a logaritmic rather than in a linear scale? In its current form, only the major concentration pikes can be identified from the figure, while potential differences in "background" concentration

between the different regions are very difficult to see.

The authors could explain a bit more why and how the T/B ratio can be used as an indicator of chemical ageing. Does the chemical activity of toluene mentioned in the text refer to its OH-reactivity? Is OH the only important oxidant for benzene and toluene, and if not, what does this mean in terms of chemical ageing?

I do not understand the meaning of statement "with similar relation for benzene" following Eq. 1.

The same sentence is repeated starting from lines 13 and 17 on page 8. Concerning the follow up of the latter sentence, one cannot infer from this information that benzene emission from motor vehicle exhaust is 25% lower than toluene emission. The logic here is incorrect!

Section 3.2

In the first sentence of page 9, do you mean "Observations in several locations have reported. . .."? If yes, then also the beginning of the next sentence need to be modified: "Contrary to these observations, no . . .".

A high pollution level itself cannot be a reason for the lack of observed diurnal cycle because, in principle, also high concentrations could be relatively evenly distributed. I would rather thing that a lack of strong local pollution sources (or lack of very high concentrations above the mixed layer) would be the reason.

The times are not usually given in a.m. or p.m (rather 04:00 and 23:00)

Section 3.3

I do not think "meaningful correlation" is proper statistical language. Furthermore, I am not confident that R=0.6 can be considered as a high correlation.

Section 4

[Figure]

What is meant by "one-time maximum permissible concentration"? Also mentioned in section 3.1 and in abstract.

Can the statement made on lines 20-23 considered general, as indicated here? These results are based on few measurement data points, so one would expect somewhat different numbers at some other time when travelling the same measurement route.

There are a few sentences that need to be re-written to make the text more understandable for the reader. A list of these sentences is given below.

Page 2, lines 15-16: Although. . .as well.

Page 6, lines 7-10: Although . . . scale.

Page 6, lines 23-25: The observed. . .power plants.

Page 7, lines 26-27: These values are. . .

Page 12, lines 12-14: These are . . . correspondingly.

Finally, there are a number of minor grammatical issues that need to be corrected. Below is a list of suggestions for such corrections:

p 2, l 17-18: . . .carried out around the world (e.g. Keymeulen. . ..)

p 3, l 3: meteorology – > meteorological quantities

p 3, l 11: . . . previously in Krutzen et al. (1996) and Elansky et al. (2000).

p 3, l 18: . . . as well as by complex. . .

p 3, l 23: . . . of the order of one week, which. . ..

p 3, l 26: delete "changing each other"

p 3, l 31: . . . clearly seen in Fig. 1 where. . .

p 3, l 34: According to Fig. 1, the. . ..

p 4, l 2: scale – > scales

p 4, l 9: VOC concentrations were measured. . ..

p 4, l 16: . . . detection limits . . . VOCs are. . .

p 4, line 21: . . . Fortner et al., 2009). However, isoprene has been found to be the dominant. . .

p 4, l 26: . . . an ALHA. . .

p 4, l 29: does not

p 5, l 1: The CO . . . with a TE48. . .

p 5, l 4: The SO2. . .. with an APSA. . .

p 5, l 6-7: . . .a TE42C. . . a M200AU. . .

p 5, l 8: . . .0.05 ppb, which makes it possible to measure so-called. . .

p 5, l 26: . . .NOx concentrations and by an increase in the NO/NO2 ratio. . .

p 5, l 27: is – > was

p 5, l 30: . . .), being safely. . .

p 6, l 4: A statistical. . .

p 6, l 5: are – > were

p 6, l 5: As discussed above, . . .

p 6, l 11: are – > were

p 6, l 16: ..towns are located. . .

p 6, l 29: . . . benzene content.

p 7, l 4: divided – > divided

p 7, l 8: are – > were

p 7, l 13: farther – > further

p 7, l 17: utilize – > utilized

p 7, l 19: latter being . . . chosen based on. . .

p 7, l 25: . . .reaching the values of 45.6 and 36.5 ppb, respectively.

p 7, l 28: . . . a robust. . .

p7, l 30: . . .were in the ranges of 1.8- . . ..

p 7, l 32: . . . were found in all the regions, suggesting other important. . .

p 8, l 2: are – > were

p 8, l 13: . . .exhausts were responsible for. . .

p 8, l 20: . . .exhaust was the most significant. . .

p 8, l 24-25: However, toluene levels tended to be lower than those reported in earlier publications.

p 9, l 10: does – > did

p 9. l 13: We found that the surface concentrations . . . were, in general, notably higher in urban. . .

p 9, l 16-17: A closer examination showed that these events were. . .

p 9, l 27: . . . was very poor, except for . . .., the diversity. . .

p 9, l 31: . . .selected, for which. . ..

p 10, l 1: are – > were

p 10, l 5: . . ., which was also confirmed by the high T/B ratios. . .

p 10, l 6: show – > indicate

p 10, l 8-9: . . .was found to be a main pollution source, even though the contribution from industrial source was also important, as seen from the significantly. . .

p 10, l 10: . . . in the medium- . . . were close to. . .

p 10, l 12: is – > was

p 10, l 13: . . .using the TROICA data due to the very limited number of observations collected. . . .

p 10, l 19: based on

p 10, l 22: is – > was

p 10, l 25: As mentioned above, . . .

p 10, l 28: employed

p 11, l 2: . . .having the dimension of. . ., respectively, „,

p 11, l 7: We calculated . . . based on. . .

p 11, l 12: is – > was

p 11, l 13-14: . . .abundances of isoprene . . .and its exceptionally. . . .

p 12, l 11: are – > were

p 12, l 14: Vehicle emissions constitute the major. . .in these regions, with typical. . .

p 12, l 16: are – > were

---

## Author Comment (AC2) · 13 Feb 2017

The authors thank the anonymous referee #2 for the constructive comments and corrections. Our response to all comments is given below.

1. Have the authors considered plotting the concentrations in Figure 3 in a logarithmic rather than in a linear scale? In its current form, only the major concentration pikes can be identified from the figure, while potential differences in "background" concentration C1 ACPD Interactive comment Printer-friendly version Discussion paper between the different regions are very difficult to see.

Reply: Thank you for this recommendation. We plotted the concentrations in a logarithmic scale.

2. The authors could explain a bit more why and how the T/B ratio can be used as an indicator of chemical ageing. Does the chemical activity of toluene mentioned in the text refer to its OH-reactivity? Is OH the only important oxidant for benzene and toluene, and if not, what does this mean in terms of chemical ageing?

Reply: The chemical activity of toluene mentioned in the text certainly refers to its OH-reactivity. OH-radicals are the main oxidants of all VOCs in the atmosphere, especially at the daytime. Toluene has a shorter atmospheric lifetime than benzene due to its faster photochemical removal by OH (rate constants of 1.23 âČř 10-12 and 5.96 âČř 10-12 cm3 molecule-1s-1 for benzene and toluene, respectively, at 298K by Atkinson and Arey, 2003). Thus, as toluene is more rapidly removed by oxidation, the T/B ratio decreases as air is transported over longer distances away from the pollution source. The chemical oxidation of benzene and toluene by NO3 and O3 is very weak (k âČř 10-17 cm3 molecule-1 s-1 and k âČř 10-18 cm3 molecule-1 s-1, respectively for NO3 and O3) with faster toluene oxidation comparing with benzene as well.

3. I do not understand the meaning of statement "with similar relation for benzene" following Eq. 1.

Reply: Agreed. It is corrected.

4. The same sentence is repeated starting from lines 13 and 17 on page 8. Concerning the follow up of the latter sentence, one cannot infer from this information that benzene emission from motor vehicle exhaust is 25% lower than toluene emission. The logic here is incorrect!

Reply: Agreed. It is corrected.

5. In the first sentence of page 9, do you mean "Observations in several locations have reported. . .."? If yes, then also the beginning of the next sentence need to be modified: "Contrary to these observations, no . . .".

A high pollution level itself cannot be a reason for the lack of observed diurnal cycle because, in principle, also high concentrations could be relatively evenly distributed. I would rather thing that a lack of strong local pollution sources (or lack of very high concentrations above the mixed layer) would be the reason.

The times are not usually given in a.m. or p.m (rather 04:00 and 23:00)

Reply: Agreed. It is corrected.

6. I do not think "meaningful correlation" is proper statistical language. Furthermore, I am not confident that R=0.6 can be considered as a high correlation. Reply: R > 0.5 may be an evidence for (potentially) strong relationship (R= 0.5 to 1.0) between the compounds observed. Some terminological corrections are included in the text (Section 3.3).

7. What is meant by "one-time maximum permissible concentration"? Also mentioned in section 3.1 and in abstract.

Reply: We meant a maximum concentration of the pollutant, short-term exposure of which (within 20 minutes) does not cause negative human effects. "Short-term exposure limit" would possibly be better. The values of such limits for benzene and toluene are presented according to the Russian regulations. Corresponding corrections are added in the text.

8. Can the statement made on lines 20-23 considered general, as indicated here? These results are based on few measurement data points, so one would expect somewhat different numbers at some other time when travelling the same measurement route. Reply: Agreed. It is corrected.

9. There are a few sentences that need to be re-written to make the text more understandable for the reader. A list of these sentences is given below.

Reply: Agreed. We tried to make them more understandable.
10. Finally, there are a number of minor grammatical issues that need to be corrected. Below is a list of suggestions for such corrections:

Reply: Agreed. We thank the referee for the corrections. All of them were included in the text.

Please also note the supplement to this comment:
http://www.atmos-chem-phys-discuss.net/acp-2016-858/acp-2016-858-AC2-supplement.pdf

[Figure]

[Figure]

[Figure]

**Fig. 1.**

**b)**

Fig. 2.

**Supplement:**

[revised manuscript text omitted]

---

## Author Response (AR2)

The authors thank the anonymous referee #2 for the minor but important corrections. All corrections are accepted and included in the text, and highlighted with yellow color.